# Analysis of the Functioning of Families of Children with Autism Spectrum Disorder: A Psychometric Study of the Family APGAR Scale

**DOI:** 10.3390/ijerph20237106

**Published:** 2023-11-24

**Authors:** Laura Serrano, Esther Vela, Laura Martín

**Affiliations:** Faculty of Education and Psychology, Universidad Francisco de Vitoria, Majadahonda, 28223 Madrid, Spain; esther.vela@ufv.es (E.V.); l.martin.prof@ufv.es (L.M.)

**Keywords:** autism spectrum disorder, family, Family APGAR, family functioning, family dynamics

## Abstract

Normalised family functioning is a predictor of individual well-being. The diagnosis of a family member with autism spectrum disorder (ASD) can alter the ordinary dynamics of family systems, having a variable impact on family functioning. This research employed a non-probability convenience sampling method to gather a sample of 327 families with and without children diagnosed with ASD. This study has dual objectives: to analyse the psychometric properties of the Spanish version of the Family APGAR Scale and to compare family functioning in families with and without a child with ASD. The results reveal several favourable psychometric properties in the application of the APGAR scale within families of children with ASD. The use of the Family APGAR Scale in the selected sample confirms that the functioning of families with children with ASD can be categorized as mildly dysfunctional, attributed to the inherent challenges in caring for and raising a child with ASD. The presence of ASD within family systems presents a challenge to typical family functioning, with significant differences observed between families with and without children with ASD. This underscores the necessity of implementing effective intervention programs based on empirical evidence to improve the quality of life for individuals with ASD and their families.

## 1. Introduction

Family is generally considered the earliest and most important institution in society, the foundation of human cultural and economic life [1], as well as the most direct and significant factor in the psychosocial development of the individual [2]. Thus, family structure and dynamics of family functioning is an effective predictor of the degree of life satisfaction of its individual members [3].

Since the 1970s, various experts in the study of family systems refer to the notion of “family functioning”, particularly Olson [4], who uses this term to refer to the interrelation and connection between family members (cohesion) and the capacity of family systems to adapt to difficult circumstances (adaptability). In parallel to the generalisation of the concepts proposed by Olson, other proposed paradigms emerged based on two divergent theoretical approaches to family functioning: results-oriented family functioning, taking into consideration specific variables for each family, corresponding to the Olson Circumplex Model [4,5] and the Beavers System Model [6,7], and the second group, process-oriented family functioning, which approaches family functioning through the type of tasks each family system assumes in different stages of life [8] corresponding to McMaster’s Theory of Family Functioning [9] and Skinner’s Family Functioning Model [10].

Although a large part of the research in family functioning, based on the models indicated above, focuses on identifying patterns of optimum family functioning [11], it is practically impossible to generalise the notion of “normality” given the complexity of variables involved and the importance of the sociocultural context in which each family ecosystem functions [12]. Nevertheless, it can be affirmed that a family can be considered “functional” when it manifests an adequate capacity to adapt to changing internal and external circumstances throughout different stages of life, maintaining the appropriate degree of cohesion among members. Thus, a functional family system is one that is able to maintain its identity despite the exposure to varying degrees of stress or conflict, whereas dysfunctional families are characterised by a lack of tools or skills necessary to successfully navigate and manage stressful circumstances, presenting non-adaptive patterns of conduct and interrelations between family members [13].

The predictive factors of family dysfunction are highly varied, and it is essential to consider these factors in context. These factors may include negative or adverse circumstances for families, such as financial problems [14], chronic illness [15], and serious accidents or death [16], as well as significant or impactful events, such as moving house, changing in employment, or the birth of a child, which can be a source of stress on the family. Among the factors that have the most impact on family functioning, some form of disability of a family member has been traditionally considered an extremely significant factor influencing individual and family dynamics [17,18] and may even represent a social stigma or discredit for the family [19]. The families of individuals with disabilities often show high levels of stress, produced by the behaviour and additional needs of a family member that may be difficult to manage, or the absence of effective coping strategies or the lack of resources or family support in raising and educating the person with a disability [20]. Specifically, the diagnosis of autism spectrum disorder (ASD) of a family member can represent an even greater risk factor for family stability and functioning [21]. Severe alterations in conduct associated with ASD have been linked to dysfunctional and/or vulnerable family systems [22,23], as well as high levels of parental stress, having a direct impact on parenting strategies and the quality of family life [24,25].

Compared to ‘normotypical’ families, those dealing with disability on a daily basis have a higher probability to suffer personal burnout [26] due to the demands of dealing with disability within the family system. For families of children with ASD, the specific characteristics of ASD are directly associated with greater levels of stress and impaired family resilience compared to families dealing with other forms of disabilities (such as Down’s syndrome, developmental alterations, etc.) [27,28,29]. Family resilience [30], cohesion, and the capacity for adaptation [31], play a crucial role in addressing the challenges of living with a person with a disability [32]. It is essential that clinical, social, and educational support services develop and deploy effective intervention programs focussed on the family [33], specifically those with individuals with a disability [34], in order to address the issues mentioned above and to enhance the quality of life for not only of those with a disability but also their entire family environment [35].

The Family APGAR Scale was developed in the late 1980s when Smilkstein [36] published and validated an initial version of the instrument. The initial purpose of the instrument was clinical, aiming to provide family doctors with a panoramic overview of the structure and functioning of families, providing primary care practitioners useful information to prevent and manage problems arising within the families of their patients. The evolution of the instrument was documented in the first and subsequent publications by the author and his collaborators [36,37]. The first version of the scale, formatted with five possible responses, showed consistent and reliable psychometric properties (Cronbach’s alpha = 0.86; total correlation between responses = 0.50 to 0.65) [38], enabling the later generalisation of its use.

Over time, the effectiveness of the Family APGAR Scale continued to be analysed as its use became more widespread and generalised in different contexts, demonstrating the validity, reliability, and utility of the tool in determining the prevalence of perceived and self-reported family dysfunction. Numerous prestigious studies focussing on the functioning of family systems affected by highly diverse conditions (including chronic illnesses such as HIV, leprosy, and diabetes) have made use of the Family APGAR questionnaire [39]. The effectiveness of this tool has been demonstrated in identifying dysfunction among family systems dealing with the challenges of disability [40,41,42,43] and, more specifically, in samples of family systems in the case of autism spectrum disorder [44]. There are different versions of the scale translated into different languages and adapted to different cultures with satisfactory psychometric values, including China [45], Spain [46], Taiwan [47], the Philippines [48], Japan [49], Poland [50], Colombia [51], Iran [52], Ghana [53], and Indonesia [54], among others. Furthermore, the tool is appropriate for use with sample populations from early childhood–youth [55,56] until the later stages of adult life [52].

The present research has dual objectives: first, to evaluate the psychometric properties of the Spanish version of the Family APGAR Scale [39], administered to a sample of families of children with autism spectrum disorder (ASD), given that the APGAR scale is one of the most standard instruments in evaluating family functioning, and second, to analyse the functioning of families of children with autism spectrum disorder compared to those without.

## 2. Materials and Methods

### 2.1. Participants

This research used a non-probability convenience method to gather a sample of families of children with and without autism spectrum disorder.

The final sample of the study consisted of 327 Spanish families, of which 160 had a child with autism spectrum disorder and 167 without. The age of the families of a child with ASD ranged from 27 to 66 years of age, with a mode of 55; the children were aged between 3 and 25, with a mode of 6.

Among families of a child with autism spectrum disorder, mothers were those who most commonly responded to the questionnaire (84.6%) compared to fathers (15.2%), with only one case where another person responded. The majority of participants were married (65.58%) and, although a high percentage were employed full time (44.5%) or part time (23.2%), over half of the families did not have any form of help in caring for their child with autism spectrum disorder (53.5%), while 40.1% had the help of family members.

Among families of children without autism spectrum disorder, mothers were also the most frequent respondents (75.3%) compared to fathers (20.5%). In this case, the majority were married (67.1%) and worked full time (54.4%) or part time (20%).

### 2.2. Community Involvement

The protagonists of this research project are people with autism spectrum disorder and their family environments. This study is based on the needs expressed by families with children with autism, with the main purpose of developing, a posteriori, professional intervention plans. To ensure the active participation of people with autism, their families were involved in the formulation of the research questions. As a result, the most appropriate standardized-assessment instrument was selected, and the families were asked to complete it. Thus, from the beginning, the participating families were deliberately and voluntarily involved in the research process. The participatory role of fathers and mothers with children with autism spectrum disorder was decisive for the development of this research.

On the other hand, the participation of professionals from the educational sector of students with autism spectrum disorders was also present, since they were in charge of mediating between the team of researchers and the sample of families with children with autism.

### 2.3. Instruments

This research made use of the Family APGAR Scale, administered along with a series of sociodemographic questions to collect information about the respondents.

Family APGAR Scale [36]: This instrument was developed to analyse and understand the functioning of a family system in a particular moment as perceived by one of its members. The APGAR scale is composed of five items that evaluate the dimensions of adaptation, partnership, growth, affect, and resolve within a family. There is a Spanish version of the scale, validated by Bellón et al. [46], and the interclass correlation coefficient of the 5 items is above 0.55 and has a Cronbach’s alpha of 0.84. The questionnaire is simple to use, administered individually, and answered personally by each participant. There are five response options on a scale of 0 to 4, where 0—Never; 1—Almost never; 2—Sometimes; 3—Almost always; and 4—Always.

### 2.4. Procedure

The first step in this research was to create the study sample. Families of children with autism spectrum disorder were contacted through special education schools within the Community of Madrid, Spain; specifically, families of children enrolled in these schools diagnosed with autism spectrum disorder. In the case of families with children not diagnosed with autism spectrum disorder, their participation was requested through ordinary schools within the Community of Madrid. In both cases, the schools that agreed to participate in this study were sent the questionnaire on paper, in a sealed envelope, to be delivered to the families in the students’ school bags. After two weeks, the completed questionnaires were collected from the schools.

### 2.5. Data Analysis

To achieve the objectives of this research, an initial analysis was made of criterion normality (Kolgomorov–Smirnov), which showed that the sample did not meet the criteria for normality (*p* < 0.05), and thus, non-parametric testing was conducted to analyse the data.

Furthermore, an analysis was made to verify the reliability of the instrument (Cronbach’s alpha), as well as confirmatory analysis. A descriptive and correlational analysis was also carried out on the variable of the study. Finally, in order to obtain evidence of criterion validity regarding the relationship between family functioning based on the presence or absence of ASD, the Spearman’s Rho test was administered.

The data were analysed using the IBM SPSS statistics software, version 29.0.

## 3. Results

### 3.1. Construct Reliability and Validity

Reliability testing confirmed that the instrument is apt for the study sample, families of children with ASD, with a Cronbach’s alpha of 0.91, considered excellent.

The confirmatory factor analysis indicates the model has an adequate goodness of fit according to the thresholds established by the authors (0.99 for CFI and TLI and 0.05 for RMSEA), with scores of CFI = 0.999, TLI = 0.997, and RMSEA = 0.026.

### 3.2. Results of the Descriptive Analysis

The following is a descriptive analysis of the results for the Family APGAR Scale for families of children with and without autism spectrum disorder. The following scale has been used in evaluating the scores:Normal: 17–20Slight dysfunction: 16–13Moderate dysfunction: 12–10Severe dysfunction: <9

In Table 1, the results of the descriptive analyses conducted are presented, both for each of the items and for the complete Family APGAR Scale (“Total APGAR”). Taking into account that the scores for each of the items range from 0 to 4, it can be determined that, in all cases, both groups of families obtained scores above the mean (placing them closer to the normal value). It is important to note that families with a child without autism spectrum disorder (ASD) obtained higher scores compared to families with a child with ASD, both for all the items and for the total score of the scale.

More specifically, it can be observed that families with a child with ASD obtained the highest score on item 3 (mean = 3.10), related to “Growth”. This indicates that these family systems feel adequately supported by their environment, as reflected through a proper perception of their individual and personal growth. In contrast, the lowest score obtained by the sample of families with a child with ASD was on item 1 (mean = 2.84), related to “Adaptation”. In this regard, it is evident that these families demonstrate a low ability to effectively cope with difficulties or stressful situations arising from their child’s autism spectrum disorder. On the other hand, families with children without ASD obtained the highest scores and, therefore, scores closer to the normal value on item 5 (mean = 3.42), related to “Resolve”, which is understood as an indicator of a high capacity to meet the physical and emotional needs of other family members. The lowest score, expressed by the sample of families with children without ASD, was observed on item 3, “Growth” (mean = 3.25), related to their individual development within the family system.

Finally, the “Total APGAR” item refers to the overall score of the scale, providing a comprehensive view of family functioning based on the analysis of scores obtained in the 5 items that make up the scale. In this regard, the data obtained demonstrate that families with a child with ASD exhibit a slight family dysfunction (mean = 15.17) in comparison to families with a child without ASD, for whom the data indicate that they are closer to the criterion of normality (mean = 16.60).

Below, the descriptive analyses of the sample of families with a child with autism spectrum disorder (ASD) are presented, categorized by the degree of severity (mild, moderate, or severe). The conducted analyses provide a profound understanding of the extent and nature of family functioning in relation to the level of manifestation and impact of symptoms associated with the diagnosed child’s autism spectrum disorder.

As can be observed in Table 2, in all cases, the means of the scores for each of the items that make up the Family APGAR Scale, as well as the results obtained in the overall score of the Family APGAR Scale (“Total APGAR”), are above the average. This occurs because responses are measured on a Likert-type scale, ranging from 0 to 4 points.

Specifically, for families with a child with a mild degree of autism spectrum disorder (ASD), it can be observed that the factors in which the highest scores are obtained are “Growth” and “Affect” (mean = 3.04). In contrast, the lowest scores are related to the “Adaptation” item (mean = 2.79). In the case of families with a child with a moderate degree of ASD, the highest scores are obtained in the “Partnership” item (mean = 3.13), while the lowest scores are found in the “Adaptation” item (mean = 2.92). Finally, families with a child with a severe degree of ASD obtained the highest score in the “Resolve” factor (mean = 3.26), while the lowest score is observed in the “Adaptation” item (mean = 2.89). 

### 3.3. Analysis of the Inferential Study

Finally, the results are displayed below in order to determine if there are any statistically significant differences between families of children with ASD and those without. And within the families that have a child with ASD, are there differences within the group based on the degree of severity.

The results from Table 3 allow us to understand whether there are statistically significant differences between families with a child with autism spectrum disorder (ASD) and those without ASD, both for each of the items comprising the Family APGAR Scale and for the total scores of the scale. The *p*-value demonstrates that there are statistically significant differences in all the items of the scale (*p* < 0.05), except for the third item, “Growth” (*p* > 0.115), which refers to the sense of self-realization achieved with the help and support of one’s own family.

This implies that in the analysis of the overall “Total APGAR” scale, statistically significant differences are observed between the two family groups. This means that, despite finding similar scores in the descriptive analyses for the means of both family groups (with ASD mean = 15.17; without ASD mean = 16.60), the Mann–Whitney U test indicates that there are indeed significant differences between the two family groups (*p* = 0.005).

With the aim of verifying the use of the family functioning scale for families with and without a child with ASD, the Spearman’s Rho test was applied. The data obtained indicate a low (Rho = 0.184) but significant (*p* < 0.05) correlation between family functioning and the presence of ASD within the family context.

Finally, Table 4 displays the results obtained after applying the Kruskal–Wallis test for the sample of families with a child with autism spectrum disorder (ASD). The purpose of this analysis is to verify the presence of possible statistically significant differences based on the degree of severity of the child’s autism spectrum disorder (mild, moderate, or severe).

As can be observed in Table 4, there are no statistically significant differences within families with a child with ASD based on the degree of severity, as for both, in each of the items and in the overall result of the complete scale, *p* > 0.05 is obtained.

## 4. Discussion

This research had two key objectives: first, to analyse the psychometric properties of the Spanish version of the Family APGAR Scale and second, to explore the functioning of families of children with ASD compared to those without.

Regarding the first objective, the APGAR scale proved to be a reliable tool (0.91) for the analysis of family dysfunction in the case of families of children with ASD. These results are in line with the findings of other studies using the same scale applied to a wide range of different samples, such as university students or the elderly [52,53,54,55,56,57], as well as families with members diagnosed with autism spectrum disorder [43,44]. In all cases, the internal consistency of the tool was considered good to excellent.

Regarding the second research objective, scores were above the mean for all items, especially the item “Growth”, which had the highest scores and refer to personal and family development and the feeling of self-fulfilment thanks to the mutual support of family members.

These findings are in line with the results of other studies, such as that by the authors of [58], who associate health with self-fulfilment and well-being, even in adverse circumstances. This study is directly linked to the sample analysed in the present study. Given that families of children with ASD face numerous challenges and difficulties associated with the care of a child with ASD, not only at the moment of diagnosis but over the course of an entire lifetime, requiring that families “are fortifying themselves and develop the capacity to find fulfilment both for themselves and for their children” [59] (p. 54).

In achieving this self-fulfilment, the emotional support of a spouse is essential, as found in the studies of family systems by Sumalavia and Almenara [60] or García-López et al. [61], which highlight the importance of the mutual support of spouses in maintaining a positive relationship and successful adaptation to adverse circumstances.

Although the scores were above the mean, there were statistically significant differences between the groups in all items, with the exception of item 3, “Growth”. This may be due to the fact that, regardless of the difficulties faced by family systems, the family continues to be the entity which most directly influences the psychosocial development and well-being of each member [2].

Of particular note are the results obtained for item 4, “Affect”. Although there were differences between the groups, these differences were at the limit of significance. The families of children with a disability are more likely to suffer greater degrees of personal burnout [26], which may influence the relationships between family members.

Additionally, the lowest scores were found for the dimension “Adaptation”, referring to the ability to adapt and overcome adversity by deploying internal and external resources. Although the scores were lowest for this dimension, they remained above the mean. In the case of families of children with ASD, this is a positive result, considering that it indicates that families have sufficient coping mechanisms, understood as the ability of family members to deploy available resources to face situations of stress [62], as well as the resources necessary to adapt to adversity [63].

The deployment of coping strategies and resources is directly associated with resilience, that is, the capacity of individuals to adapt to adversity [64]. According to Fínez et al. [65], resilient individuals do not undergo a period of recovery before difficulties but rather are able to maintain an equilibrium in family functioning. The scores for this item indicate that families of children with ASD have developed a high degree of resilience, most likely due to the prevalence of stressful situations or difficulties they must face [30].

The scores for families of children with ASD indicate that these families have more resources and information, achieving a greater degree of acceptance of disability within the family. As the study by Serrano [66] concludes, “the understanding and general view parents have of the disability of their children are considered positive, resulting in the sensation of a normal family functioning” (p. 23).

## 5. Conclusions

This research offers a deeper understanding of the functioning of families with children with autism spectrum disorder (ASD) through a comparative analysis conducted between the dynamics existing in families with children with and without ASD. To obtain the results, the Family APGAR Scale was utilized, which demonstrated appropriate and consistent psychometric properties, confirming the reliability of the tool for the sample used.

The findings suggest that, although families face numerous challenges, they exhibit a remarkable capacity to function and adapt positively to adversity. Given the high variability in the manifestation of the autism spectrum, we delved into the analysis of how the degree of ASD severity in the individual influenced family functioning. The data indicate that there are no significant differences in family functioning based on the degree of ASD severity. However, differences were identified between the functioning of families with and without children with ASD, showing that families with children without ASD tend to exhibit functioning closer to normalcy compared to families with children with ASD, which tend to exhibit dysfunctional dynamics. The results highlight the importance of continued support and resources for these families, as well as recognition of their strength and well-being.

The present study has several limitations, the main one being a limited sample size, consisting of fewer than 200 participants for each of the sample subgroups analysed. The sample size, combined with the use of non-parametric tests, hinders the ability to generalize the results, understanding that the conclusions of this research specifically pertain to the analysis of the obtained results. The limitations inherent in this research underscore the need to further examine the specific factors that contribute to positive family functioning, particularly in families of children with autism spectrum disorder. Greater understanding will permit the development of more effective family intervention strategies and programs aimed at the specific aspects where greater help is most needed.

## Figures and Tables

**Table 1 ijerph-20-07106-t001:** Descriptive results of the sample of families of children with and without ASD.

	ASD	N	Median	Mean	Standard Dev.
Adaptation	Yes	159	3.00	2.84	1.256
	No	166	4.00	3.31	1.089
Partnership	Yes	159	3.00	3.00	0.994
	No	166	4.00	3.33	0.834
Growth	Yes	156	3.00	3.10	1.021
	No	166	4.00	3.25	0.970
Affect	Yes	156	3.00	3.06	1.045
	No	167	4.00	3.28	0.980
Resolve	Yes	157	3.00	3.07	0.907
	No	166	4.00	3.42	0.764
APGAR Total	Yes	151	15.00	15.17	0.907
	No	163	18.00	16.60	3.688

**Table 2 ijerph-20-07106-t002:** Descriptive results of the sample of families of children with ASD based on the degree of severity.

	Degree of Severity	N	Median	Mean	Standard Dev.
Adaptation	Mild	73	3.00	2.79	1.258
	Moderate	52	3.00	2.92	1.234
	Severe	28	3.00	2.89	1.286
Partnership	Mild	73	3.00	2.92	0.954
	Moderate	52	3.00	3.13	1.010
	Severe	28	3.00	2.96	1.105
Growth	Mild	70	3.00	3.04	0.999
	Moderate	52	3.00	3.12	1.041
	Severe	28	3.00	3.11	1.133
Affect	Mild	70	3.00	3.04	0.924
	Moderate	52	3.00	3.06	1.127
	Severe	28	3.00	3.00	1.247
Resolve	Mild	72	3.00	3.01	0.847
	Moderate	52	3.00	3.12	0.963
	Severe	27	4.00	3.26	0.944
APGAR Total	Mild	66	15.00	15.02	4.071
	Moderate	52	16.00	15.35	4.769
	Severe	27	17.00	15.22	5.235

**Table 3 ijerph-20-07106-t003:** Inferential analysis using the Mann–Whitney U test of families with and without ASD.

	ASD	N	Mean Rank	Sum of Ranks	U	*p*
Adaptation	Yes	159	144.08	22,909.00	10,189.0	<0.001
	No	166	181.12	30,066.00		
Partnership	Yes	159	147.54	23,458.50	10,738.5	0.002
	No	166	177.81	29,516.50		
Growth	Yes	156	153.69	23,975.00	11,729.0	0.115
	No	166	168.84	28,028.00		
Affect	Yes	156	151.90	23,697.00	11,451.0	0.042
	No	167	171.43	28,629.00		
Resolve	Yes	157	143.86	22,585.50	10,182.500	<0.001
	No	166	179.16	29,740.50		
APGAR Total	Yes	151	142.72	21,551.00	10,075.000	0.005
	No	163	171.19	27,904.00		

**Table 4 ijerph-20-07106-t004:** Inferential analysis using the Kruskal–Wallis test in families with children with ASD based on the degree of severity.

	Degree of Severity	N	Average Rank	H Kruskall–Wallis	*p*
Adaptation	Mild	73	74.61	0.454	0.797
	Moderate	52	79.38		
	Severe	28	78.82		
Partnership	Mild	73	72.32	2.264	0.322
	Moderate	52	83.78		
	Severe	28	76.63		
Growth	Mild	70	72.99	0.508	0.776
	Moderate	52	77.39		
	Severe	28	78.27		
Affect	Mild	70	73.47	0.326	0.850
	Moderate	52	77.48		
	Severe	28	76.89		
Resolve	Mild	72	70.90	2.701	0.259
	Moderate	52	78.16		
	Severe	27	85.43		
APGAR Total	Mild	66	69.05	1.090	0.580
	Moderate	52	76.28		
	Severe	27	76.33		

## Data Availability

The data presented in this study are available on request from the corresponding author.

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
