# Peer review of "Analysis of the Functioning of Families of Children with Autism Spectrum Disorder: A Psychometric Study of the Family APGAR Scale"

_ijerph, 2023, doi:10.3390/ijerph20237106_

Round 1

Reviewer 1 Report

Comments and Suggestions for Authors

I consider that in the conclusions it should be considered that the APGAR scale is a reliable tool for the analysis of family functioning because it is one of the objectives of the research carried out.

Author Response

First and foremost, we would like to express our gratitude for the reviewer's dedication to thoroughly analyze the content of this paper. Their critical insights have allowed us to deeply review the contents presented herein.

Regarding the conclusions, we would like to confirm that we have proceeded to detail the fact that the Family APGAR Scale is a reliable tool to analyze the functioning of families with a member diagnosed with Autism Spectrum Disorder (ASD).

We hope that the corrections made meet the reviewer's expectations. We deeply appreciate the time and dedication invested in reviewing the paper.

Reviewer 2 Report

Comments and Suggestions for Authors

  The author's work exploring the psychometric indicator of the Family Apgar scale in families with autistic children is meaningful and will help the study in this area. Well, I am afraid the author should do more work to improve the article.

1. A larger sample size will benefit the data analysis and make the result more stable and representative. The sample should ideally encompass at least 200 participants for each group.

2. In section 3.2, Line 204-208, only criteria scores of sublevels of the Family Apgar scale were given, and it is strongly recommended to provide specific distributions of each sublevel for both groups to test the distribution difference from the two groups.

3. As the intention of the article was to test the psychometric characteristics of the Family Apgar scale in families with autistic children, test-retest reliability and criterion-related validity were needed to verify the use of the scale in the test population(ASD family).

4.  The Family Apgar scale only has five items, each representing one subfactor; it's not proper to compare each item score between two groups even though the author uses a Non-parametric test. A parametric test on full-scale scores with supplemented samples should be better.

4. For the non-parametric test, table 4,  first raw, "mean ranks / sum of ranks" rather than "mean ranges / sum of ranges." The meticulous use of accurate terminology is essential for conveying research methods and outcomes with precision.

5. Additionally, there are some grammar/spelling that need to be corrected. For example, in line 232,  "to key objectives" was wrong of " two key objectives." A careful check is needed.

Author Response

  1. First and foremost, we express our gratitude to the reviewer for their comments, acknowledging one of the primary limitations of the conducted study, which is the sample size. In the case at hand, it is important to emphasize that, despite being fully aware of the necessity for a representative sample size to conduct high-quality research with generalizable results, the sample used for this study is exclusively comprised of families with individuals diagnosed with Autism Spectrum Disorder (ASD). Consequently, access to a larger representative sample than that employed in the present study is contingent upon the willingness of individuals with a diagnosis of ASD to actively engage as study participants. However, recognizing that this is a fundamental limitation of the current research, we have duly documented the existence of this limitation in the 'Conclusions' section of the main body of the paper.
  2. The Family APGAR Scale is a multifactorial tool for assessing family functioning, consisting of 5 items with a Likert-type response format, ranging from 0 to 4 points. To evaluate the degree of adjustment in a family system's functioning, it is necessary to calculate the mean of the responses obtained for the 5 items that make up the scale. The total score (see "Total APGAR" category in Table 3) is the value that indicates the level of family functioning or dysfunction, taking into account the following values:

    - Normal: 17-20

    - Slight dysfunction: 16-13

    - Moderate dysfunction: 12-10

    - Severe dysfunction: <9.

    In order to explain the functioning of the scale scores and their interpretation, an explanatory paragraph has been included in the main text, following Table 3, which details the obtained results and their interpretation.

    Furthermore, for the purpose of providing a more in-depth explanation, Table 4 presents data related to the results obtained in the research after administering the Family APGAR Scale to the study's participant sample. The results include data for each of the scale's items and the overall score for each of the two subgroups within the study sample (families with children with ASD and families with children without ASD). In this regard, the data analysis reveals significant differences between both subgroup samples in all items, except for item 3 ("Growth"). To elucidate this information, an explanatory paragraph has been added after Table 4, detailing the pertinent results analysis.

  3. In this research, the test-retest procedure could not be conducted as the questionnaires were administered to the sample on a single occasion, analyzing the reality of the study participants at a specific point in time. To ensure criterion validity for the selected sample, the results obtained after applying the Spearman's Rho test have been incorporated. The explanation of the administration of this test has been included within the "Data Analysis" section, while the results thereof have been elaborated on in the "Results" section, following the explanation of the inferential analyses.
  4. While we acknowledge the limitation posed by the use of non-parametric tests, the results of the Kolmogorov-Smirnov test, which yielded a score of p<.05, indicate the necessity of applying these types of tests. An explanation for the use of this test has been included in the "Conclusions" section of the main text of the paper, acknowledging it as a limitation inherent to the conducted research.
  5. Firstly, we would like to apologize for not meticulously using the terminology regarding the research methods used. To solve this problem, a review of the body of the text of the paper has been carried out, placing special emphasis on verifying that the terminology used was appropriate. We hope that the text lives up to what was requested, hoping that you can excuse any errors in terminological use and with the aim, on our part, of correcting any possible errors that may exist after the initial writing of the paper. Likewise, it is confirmed that the commented error has been solved and the expression “average range/sum of ranks” has been replaced instead of “average range/sum of ranks”.
  6. Along the same lines as in the previous comment, excuse the typos that exist in the body of the text of the paper. The comments have invited us to do an exhaustive review of the text to correct any existing grammatical, spelling and/or syntax errors. We hope all of this has been resolved correctly.

 We deeply appreciate the comments received and hope that our response meets the expectations of the reviewer. Thank you very much.

Reviewer 3 Report

Comments and Suggestions for Authors

Thank you for the opportunity to review this text. The topic is interesting, but the presented work has significant shortcomings.

Why was it necessary to mention the validation of already verified parameters (Bellón, J.; Delgado, A.; Luna, J.; Lardelli, P. Validez y verifiability del questionnaire de funcio familiar Family APGAR. Atención 410 Primaria, 1996, 8, 289-295). Was the goal of the work to validate the tool for a category of families with children with ASD? Autism is a spectrum disorder, which means that it can manifest very differently in each person because it encompasses a wide range of characteristics. These characteristics have a different degree and form of manifestation and thus influence the family and its functioning. If you state that ASD has a moderate effect on family dysfunction. The same applies to the control sample. Does averaging the differential burden of handicap affect results in a group with children with ASD? The large age range of the ADD sample 3–25 may have a similar effect. Equally embarrassing is the optimism of the authors in the conclusion, where they talk about "...more effective strategies and programs of family intervention...", because individualization is the basic requirement for special pedagogical and psychological intervention.

Author Response

Following the recommendations of the reviewer, various modifications have been made to the main body of the paper to provide an appropriate response to them

First and foremost, it is important to note that the factorial analysis of the Family APGAR scale has been removed as it was considered redundant, given the existence of prior validations of the scale in samples similar to those used in the present research. Since the primary objective of the study was not to validate the instrument used, it is emphasized that reference should be made to the existing literature that demonstrates the existence of previous psychometric studies validating the scale. For instance, the study conducted by Bellón and colleagues (1996) is cited as an example.

Additionally, Autism Spectrum Disorder (ASD) has been considered from a broad and multidimensional perspective, acknowledging the presence of various manifestations that fall within the disorder and affect individuals variably. In this regard, both descriptive and inferential analyses have been expanded for the group of families with children aged 3 to 25 years who have a child with ASD. The objective of this expansion is to gain a deeper understanding of whether the severity of the child's ASD (mild, moderate, or severe) can influence family functioning. The results of these analyses have been included in the paper's main text, specifically in Tables 2 and 4.

Furthermore, the authors recognize the limitation of expressing optimism about the potential generalizability of the results obtained in the present research to the reality of other families with children with ASD. Therefore, the inability to generalize the obtained results is acknowledged as one of the primary limitations of this study. In alignment with the reviewer's observation, there is full agreement with the need to base intervention programs on the Person and Family-Centered Approach, always considering the principle of individualization as a fundamental and essential requirement for implementing genuinely successful and effective interventions.

We hope that the corrections made meet the reviewer's expectations. We deeply appreciate the time and dedication invested in reviewing the paper.

Reviewer 4 Report

Comments and Suggestions for Authors  

Thank you for the opportunity to review the article.   The authors chose an interesting and current topic. The theoretical part is written quite well. However, I have a few comments about the research: I find the first part of the results unnecessary and inappropriate. Why did the authors factor analysis? This is a redundant step. Furthermore, these results are interpreted inappropriately.   A deeper analysis of the relations between the two research groups is missing, which was actually the purpose of this research. The conclusions are very brief and do not answer the questions posed.   I recommend reworking the entire research part of the article: - remove the factor analysis - Carry out a quality comparative study with appropriate conclusions

Author Response

First and foremost, we would like to express our deep gratitude for the commitment demonstrated by the reviewer in the review of the proposed paper. All the comments provided have been of great value in critically revising the work and enhancing its quality.

In accordance with the reviewer's recommendation, we have proceeded to eliminate the factorial analysis of the Family APGAR scale as it was considered redundant, given the existence of previous validations of the scale in samples similar to those used in the present research. Since the primary objective of the study was not to validate the instrument used, we emphasize the importance of referencing bibliographic sources that demonstrate the existence of prior psychometric studies that have conducted the scale's validation, such as the study conducted by Bellón and colleagues (1996). It is important to note that the confirmatory analysis has been retained for the purpose of demonstrating the reliability of the model used for the study sample, which comprises families with and without Autism Spectrum Disorder (ASD).

In the same line of work, it is noted that we have expanded the descriptive and inferential analyses in families with a child with ASD, considering the degree of severity (mild, moderate, severe) as a variable that can significantly influence family functioning. The results of these analyses have been included in the main text of the paper, presented in Tables 2 and 4.

We hope that the corrections made meet the reviewer's expectations. We deeply appreciate the time and dedication invested in reviewing the paper.

Round 2

Reviewer 3 Report

Comments and Suggestions for Authors

The authors have improved the quality of the text in all aspects and after minor editing I recommend it for publication.

Note: The full name Autism Spectrum Disorder appears almost fifty times in the text, although the abbreviation ASD is introduced at the beginning, already in the abstract. I recommend editing.

Reviewer 4 Report

Comments and Suggestions for Authors

The authors revised the text based on recommendations.